# Molecular Epidemiological Investigation of Piroplasms and Anaplasmataceae Bacteria in Egyptian Domestic Animals and Associated Ticks

**DOI:** 10.3390/pathogens11101194

**Published:** 2022-10-16

**Authors:** Sobhy Abdel-Shafy, Hend H. A. M. Abdullah, Mohamed K. Elbayoumy, Bassma S. M. Elsawy, Mohamed R. Hassan, Mona S. Mahmoud, Ahmed G. Hegazi, Eman H. Abdel-Rahman

**Affiliations:** 1Department of Parasitology and Animal Diseases, Veterinary Research Institute, National Research Centre, Donkki, Giza 12622, Egypt; 2Department of Zoonotic Diseases, Veterinary Research Institute, National Research Centre, Donkki, Giza 12622, Egypt

**Keywords:** *Anaplasma*, *Babesia*, Egypt, *Ehrlichia*, phylogeny, *Rhipicephalus annulatus*, *Theileria*

## Abstract

Piroplasmosis and anaplasmosis are serious tick-borne diseases (TBDs) that are concerning for the public and animal health. This study aimed to detect the molecular prevalence and epidemiological risk factors of *Piroplasma* and *Anaplasma* species in animal hosts and their associated ticks in Egypt. A total of 234 blood samples and 95 adult ticks were collected from animal hosts (112 cattle, 38 sheep, 28 goats, 26 buffaloes, 22 donkeys, and 8 horses) from six provinces of Egypt (AL-Faiyum, AL-Giza, Beni-Suef, Al-Minufia, Al-Beheira, and Matruh). Blood and tick samples were investigated by polymerase chain reaction coupled with sequencing targeting 18S and 16S RNA genes for *Piroplasma* and anaplasmataceae, respectively. Statistical analysis was conducted on the potential epidemiological factors. Of the 234 animals examined, 54 (23.08%) were positive for pathogens DNA distributed among the six provinces, where 10 (4.27%) were positive for *Piroplasma*, 44 (18.80%) for anaplasmataceae, and 5 (2.14%) were co-infected. Co-infections were observed only in cattle as *Theileria annulata* and *Anaplasma marginale* plus *Babesia bigemina*, *A. marginale* plus *B. bigemina*, and *T. annulata* plus *B. bigemina*. Piroplasmosis was recorded in cattle, with significant differences between their prevalence in their tick infestation factors. Animal species, age, and tick infestation were the potential risk factors for anaplasmosis. All ticks were free from piroplasms, but they revealed high prevalence rates of 72.63% (69/95) with anaplasmataceae. We identified *T. annulata*, *B. bigemina*, and *A. marginale* in cattle; *A. platys* in buffaloes; *A. marginale* and *A. ovis* in sheep; for the first time, *A. ovis* in goats; and *Ehrlichia* sp. in *Rhipicephalus* *annulatus* ticks. Our findings confirm the significant prevalence of piroplasmosis and anaplasmosis among subclinical and carrier animals in Egypt, highlighting the importance of the government developing policies to improve animal and public health security.

## 1. Introduction

Tick-borne diseases (TBDs) cause serious health concerns worldwide. TBDs generate severe health problems in Egypt, particularly in exotic and cross-bred cattle, affecting animals’ well-being and the livelihoods of their owners [1].

Piroplasms are apicomplexan tick-borne parasites that are found worldwide. They cause piroplasmosis (theileriosis and babesiosis) in Vertebrata and are hence medically and economically significant [2]. In Egypt, the most frequent TBDs are bovine theileriosis caused by *Theileria annulata* and bovine babesiosis caused by *Babesia bovis* and/or *Babesia bigemina* [1,3,4]. Members of the genus *Theileria* cause bovine theileriosis by acting as compulsory intercellular parasites, attacking both red and white blood cells of the hosts with their sporozoites, and reversible transformation to an uncontrolled proliferative state by the schizont stage, resulting in anemia, fever, leucopenia, and lymphoproliferative disease. The causal agent *T. annulata* infects host monocytes/macrophages and B lymphocytes and causes tropical theileriosis (Mediterranean theileriosis) [5]. Tropical theileriosis is common in Southern Europe, Asia, North Africa, and the Middle East [6,7,8]. In addition, infection with *Babesia* causes significant economic costs in cattle, including death, production loss, lower feed intake and feed conversion efficiency, abortion, tick control losses, and disease prevention losses [9,10]. The most prevalent clinical manifestations linked with *Babesia* species infections in cattle include anemia, hemoglobinemia, pyrexia, and hemoglobinuria [10,11].

Anaplasmosis is a non-contagious infectious bacterial disease caused by the family Anaplasmataceae (*Anaplasma* spp. and *Ehrlichia* spp.). *Anaplasma marginale* is found worldwide and affects erythrocytes. Affected animals remain carriers for the rest of their lives. Anaplasmosis causes significant economic losses in developing countries owing to its endemic nature [4,12]. The main signs of bovine anaplasmosis are quite varied, ranging from subclinical chronic infections to severe instances accompanied by fever, hemolytic anemia, abortions, productivity losses, and mortality [12,13]. Bovine anaplasmosis, which affects various animals, is endemic in tropical and subtropical areas [4,8,14,15].

Remarkably, the majority of animals that recovered from clinical sickness caused by *Theileria* and *Babesia* parasites became carriers [16,17]. Furthermore, subclinical infections appeared in some animals due to their resistance to clinical piroplasmosis. Therefore, carrier and subclinical case detection is a serious step for appraising the level of risk caused by piroplasms (*Theileria* and *Babesia*) [18]. Similarly, because anaplasmosis is endemic in Egypt, herd screening for Anaplasmataceae was suggested in the absence of signs or symptoms of infection [12]. Because of the low bacteremia, detecting infection in carriers by standard microscopy techniques is difficult. Hence, the polymerase chain reaction (PCR) technique has been detecting infections with low parasitemia and/or bacteremia [19,20]. In addition, when antibodies are not yet detectable by serological methods, the PCR technique can detect pathogen DNA in the acute phase of infection [20,21]. Therefore, for the epidemiological investigation of *Theileria*, *Babesia*, *Anaplasma*, and *Ehrlichia* infection, DNA-detection techniques such as PCR assays followed by sequencing are preferred [18,22,23]. To conduct an epidemiological investigation in six Egyptian provinces, this study aimed to detect the molecular prevalence, distribution, and risk factors associated with piroplasm and Anaplasmataceae in animal hosts and their associated ticks, and analyze their phylogeny. As a result, depending on the outcomes of such surveys, hemopathogen control strategies could be optimized.

## 2. Results

### 2.1. Prevalence of Pathogens

Of the 234 animals examined, 54 apparent healthy animals (23.08%) were detected as positive for pathogens DNA, 10 (4.27%) were positive for *Piroplasma*, 44 (18.80%) were positive for Anaplasmataceae, and 5 (2.14%) were co-infected (Table 1). Generally, the overall prevalence of Anaplasmataceae was significantly higher than that of *Piroplasma* (χ^2^ = 21.407, *p* < 0.001). The prevalence of Anaplasmataceae in Al-Faiyum and Matruh was significantly higher (*p* ≤ 0.001) than that of *Piroplasma* Moreover, the prevalence of Anaplasmataceae was higher in Al-Beheira and Al-Giza than that of *Piroplasma*, but the differences between the two pathogens in these two provinces were not significant (*p* > 0.05). However, in Beni-Suef, the prevalence of *Piroplasma* was not significantly higher (*p* > 0.05) than in Anaplasmataceae (Table 1). 

### 2.2. Analysis of Epidemiological Factors

All domestic animals (buffaloes, sheep, goats, donkeys, and horses) were free from piroplasmosis, except cattle that exhibited an 8.93% (10/112) prevalence rate, whereas different animal hosts revealed variable prevalence rates with anaplasmosis. Horses were free from Anaplasmataceae DNA (Table 2). Anaplasmosis detection between animal hosts exhibited a significant difference (χ^2^ = 38.923, *p* < 0.001), where the highest prevalence was found in sheep and the lowest was in donkeys. In addition, the prevalence of Anaplasmataceae in cattle was significantly higher than that of *Piroplasma* (χ^2^ = 8.526, *p* = 0.004), whereas other animals were free from *Piroplasma* (Table 2). Regarding sex, all animals positive for *Piroplasma* were females, whereas males were free from piroplasmosis. However, the difference in the prevalence of Anaplasmataceae was insignificant between females and males. In females, the prevalence of *Piroplasma* was significantly lower than that of Anaplasmataceae (χ^2^ = 11.524, *p* = 0.001) (Table 2). For age, the prevalence rate of piroplasmosis was recorded in animals aged >1 and ≤1 year, without significant difference between them (χ^2^ = 0.400, *p* = 0.527) (Table 2). Nevertheless, the prevalence rate of anaplasmosis in animals aged >1 year was significantly higher than in those aged ≤1 year (χ^2^ = 16.67, *p* < 0.001) (Table 2). In general, Anaplasmataceae was higher than *Piroplasma* in both ages (<1 and >1 year), but the increase in Anaplasmataceae was significant in those aged >1 year (χ^2^ = 22.261, *p* < 0.001) (Table 2). Regarding tick infestation, significant differences were found between infested and non-infested animals by ticks in *Piroplasma* (χ^2^ = 11.842, *p* = 0.001) and Anaplasmataceae (χ^2^ = 12.789, *p* < 0.001). Both infested and non-infested animals in anaplasmosis revealed a higher significant rate than *Piroplasma* detection (Table 2).

### 2.3. Sequencing and Phylogenetic Analyses of Pathogen in Animal Hosts

For pathogens in animal hosts, DNA sequencing confirmed the amplification of 18S and 16S rRNA genes for *Piroplasma* and Anaplasmataceae, respectively. For *Piroplasma*, 10 cattle were detected with two species of *Theileria* and *Babesia*; *T. annulata* was found in 5 cows and *B. bigemina* in 9 cows, with 4 cows co-infected with both species (GenBank: OL909618 and OM908529; respectively) (Figure 1). By BLAST analysis, we identified a potential new genotype of *T. annulata* with 99% (361/364) similarity to the *T. annulata* detected in cattle from Italy (GenBank: MT341858). Moreover, a potential new *B. bigemina* genotype that was detected with 97% (334/345) similarity to the *B. bigemina* was detected in a tick from a buffalo in Iraq (GenBank: MH356482). The phylogenetic position of both *Piroplasma* was illustrated in Figure 1. Furthermore, BLAST analysis of the obtained sequences of Anaplasmataceae revealed that cattle and sheep were positive for two different genotypes of *A. marginale*. One genotype was derived from cattle and sheep (GenBank: OL721673 and OL721674) with 100% (431/431) similarity to *A. marginale* detected in cattle blood from Cuba (GenBank: MK804764). Another potential novel genotype of *A. marginale* was derived from one cattle (GenBank: OL721672) with 99% (430/431) similarity to the same reference. Moreover, sheep and goats were positive for *A. ovis* (GenBank: OL721675 and OL721676) with 100% (431/431) similarity to *A. ovis* detected in goat blood from China (GenBank: MG869525). Meanwhile, buffaloes were positive for *A. platys* (GenBank: OL721670) with 100% similarity to *A. platys* detected in cattle blood from China (GenBank: MF289478). The phylogenetic positions of these *Anaplasma* spp. are illustrated in Figure 2.

Finally, co-infection was detected in seven cattle that were positive for more than one tick-borne pathogen (7/112; 6.25%). Triple co-infections were observed in two cattle as *T. annulata* and *A. marginale* plus *B. bigemina* (2/112; 1.78%). In addition, five other co-infections were observed as *A. marginale* plus *B. bigemina* (3/112; 2.68%) and *T. annulata* plus *B. bigemina* (2/112; 1.78%).

### 2.4. Ticks and Associated Pathogens

Ticks were found only on cows and buffaloes. The cattle tick *Rhipicephalus annulatus* was observed on cows at all localities except Al-Minufia and beside one buffalo at Al-Faiyum (Table 3). All ticks were free from *Piroplasma* DNA, but they revealed high prevalence rates with Anaplasmataceae 72.63% (69/95) in *R. annulatus*. PCR and sequencing successfully detected DNA of Anaplasmatacae in tick samples, and the obtained good-quality sequences were identified as *Ehrlichia* sp. BLAST analysis revealed a potential novel *Ehrlichia* sp. (GenBank: OL721671) with 100% (469/469) similarity to *Ehrlichia* sp. detected in *R. microplus* from China (GenBank: AF414399). The phylogenetic position of this potentially novel *Ehrlichia* sp. was in a separate clade and clustered with other *Ehrlichia* spp. in a well-supported branch (bootstrap value 93). The phylogenetic position of this *Ehrlichia* sp. is presented in Figure 2.

## 3. Discussion

Piroplasmosis and anaplasmosis are major tick-borne diseases that infect ruminants and other mammalian species in tropical and subtropical areas [24,25]. Fever, oculo-nasal discharge, increased heart and respiratory rate, aberrant mucous membrane, and low PCV (Packed cell volume) values are all signs of acute piroplasmosis or anaplasmosis in animals, making them medically and economically important [2,24]. These signs are common, although they are not pathognomonic; animals with persistent infections can be asymptomatic carriers. Carrier animals with no clinical symptoms are thought to be a major reservoir of infection for ticks, which can spread the disease to other animals [12,18,26]. The goal of this study was to investigate *Piroplasma* and Anaplasmataceae DNA in various animal hosts (cattle, sheep, goats, buffaloes, donkeys, and horses) and their associated ticks across six provinces in Egypt. A molecular investigation was performed in blood and tick samples. Epidemiological data of each animal were gathered, and data were statistically analyzed using the χ^2^ test in SPSS.

In this study, the overall prevalence of pathogens was 23.08%, where it was 4.27% for piroplasmosis in five provinces (Al-Minufia was free from piroplasm at the time of investigation) and 18.8% for anaplasmosis in all six provinces. Since 1966, many studies have stated the endemicity of piroplasmosis and anaplasmosis in several provinces in Egypt [27,28,29]. In accordance with our results, piroplasms (*T. annulata* and *B. bigemina*) in cattle were recorded in Al-Faiyum, Al-Beheira, Al-Giza, and Beni-Suef [4,22,27,30]. To our knowledge, piroplasmosis was detected for the first time among two cows in Matruh. This finding might be attributed to the trading of live animals or transferred from neighboring provinces such as Alexandria and Al-Beheira. Moreover, Al-Minufia was free from piroplasms. This finding was possibly due to the low number of investigated animals that might not be exposed to ticks. By contrast, other studies have reported piroplasms in cattle from Al-Minufia [22,27,31]. In addition, piroplasmosis was detected in other provinces such as Gharbia [32], Port Said [33], Dakahlia [24], El-Wady El-Geded [3,4], Assiut and Kharga [34], and Qena [4]. Globally, theileriosis and babesiosis were detected in animals from different countries such as Pakistan [8], Brazil [9], Malaysia [10], and Mozambique [11]. For anaplasmosis, we detected *Anaplasma* spp. in all studied provinces, which was parallel with those previously reported in the same provinces in Egypt [1,4,28,35]. Furthermore, anaplasmosis was reported previously in other provinces rather than the investigated ones [28] such as Dakahlia and Demiatta [24,36,37], Sohag and Qena [38], El-Wady El-Geded, El Minia, and Assiut [1,4,34]. Therefore, piroplasmosis and anaplasmosis circulate between animal hosts and provinces in Egypt. Additionally, anaplasmosis spreads globally between different animal hosts [13,14].

On the basis of epidemiological factors, the prevalence rate of piroplasmosis in cattle was 8.93%, while other animal hosts were free from piroplasm. This result agreed with those of Elsify and his colleagues [22] and Abdullah and her colleagues [4], who recorded a high prevalence rate of piroplasmosis in cattle, whereas other animal hosts recorded a low prevalence rate or were free from piroplasms. For anaplasmosis, the overall prevalence rate of anaplasmosis was 18.8% in all animal species, except horses. In Egypt, several studies have recorded the endemicity of anaplasmosis in cattle [1,4,24,28,38,39], buffaloes [4,12], sheep [4,35], and donkeys [40]. However, anaplasmosis (*A. ovis*) was firstly detected in goats in Egypt. A recent study reported *Anaplasma* antibodies in goats [41]. Furthermore, the prevalence of anaplasmosis in cattle was significantly higher than that of piroplasmosis, which is in accordance with the findings of El-Ashker and his colleagues [24] and Abdullah and her colleagues [4]. Therefore, the finding of high prevalence rates of *Piroplasma* and Anaplasmatacea among apparent healthy animals with the increase in international animal trades implies the risk of emergence and re-emergence of new genotypes of pathogens from neighboring endemic countries [7,14]. Regarding sex, no significant difference was found in the prevalence rate of anaplasmosis between males and females, whereas piroplasmosis was only recorded in females, which was in agreement with the finding of Boussaadoun and his colleagues [42] in Northwest Tunisia. This finding might be attributed to the infection being linked to stress factors, such as pregnancy, parturition, and milk production [38]. Furthermore, the age of animal hosts is regarded as a significant risk factor; according to our observations, animals aged <1 year revealed insignificant higher prevalence rates of piroplasmosis than older ones. These results were in agreement with Al-Hosary and her colleagues [3], who reported a high prevalence of piroplasmosis in younger cattle. Conversely, some reports recorded a high prevalence of such diseases in older animals [27,30]. This finding may be related to the accumulation of infections, which increases protective immunity linked with immune system maturation [3]. However, a highly significant prevalence rate of anaplasmosis was recorded in older animals than in younger ones, which is in agreement with the findings of Parvizi and his colleagues [28]. Finally, a tick infestation is a fundamental risk factor for piroplasmosis and anaplasmosis. Related to our results, a highly significant prevalence rate of tick-infested animals was found in piroplasmosis and anaplasmosis than in non-infested animals. This investigation confirms the role of ticks as vectors in the spread of these diseases between animal hosts [34,40].

Regarding the phylogenetic analysis of pathogens, we identified two species of piroplasms, namely *T. annulata* and *B. bigemina*, in cattle based on the 18S rRNA gene. BLAST analysis revealed that two potential novel genotypes, *T. annulata* and *B. bigemina*, were identified (GenBank: OL909618 and OM908529, respectively). *T. annulata* and *B. bigemina* were reported in numerous studies in Egypt [3,4,24,38]. In addition, bovine theileriosis and babesiosis have been detected in countries such as Pakistan [8], Malaysia [10], the Philippines [43], and Burkina Faso [44].

With regard to Anaplasmataceae, BLAST analysis determined the two genotypes of *A. marginale* in cattle and sheep (GenBank: OL721672 and OL721674, respectively). In Egypt, *A. marginale* was reported as an endemic pathogen in cattle [1,4,24,28,39]. In accordance with our results, *A. marginale* was detected in sheep later by Abdullah and her colleagues [4]. Moreover, *A. ovis* was identified in sheep and goats (GenBank: OL721675 and OL721676). Studies have reported *A. ovis* in sheep in Egypt [4,35,45]. In Africa, other studies have identified *A. ovis* in sheep from Tunisia [46], Senegal [47], and Algeria [23]. To our knowledge, *A. ovis* has never been detected in goats in Egypt. Recently, some studies have reported goats infected with *A. ovis* in Iraq [48], Thailand [49], and Bangladesh [50]. Likewise, we found that buffaloes were positive for *A. platys* (GenBank: OL721670). According to our findings, *A. platys* was later detected in buffaloes in Egypt by Abdullah and her colleagues [4]. In parallel, a study detected *A. platys* in buffaloes in Thailand [51]. Finally, we recorded the co-infection rate in cattle (6.25%), including triple co-infections with *T. annulata*, *A. marginale* plus *B. bigemina*, and double co-infection with *A. marginale* plus *B. bigemina*, and *T. annulata* plus *B. bigemina*. Co-infections have been commonly reported in cattle [1,4,35,52,53].

Regarding pathogens’ DNA detection in ticks, we found cattle ticks (*R. annulatus*) positive for a potential novel *Ehrlichia* sp. (GenBank: OL721671), which clustered in a separate clade with other *Ehrlichia* spp. Therefore, further genetic studies are needed using species-specific primers for verifying the novelty of the family Anaplasmataceae and detecting co-infection with *Anaplasma* and *Ehrlichia* in ticks as the main vector of anaplasmosis. Recently, Abdullah and her colleagues [40] reported a new *Ehrlichia* sp. in *R. annulatus* collected from donkeys in Beni-Suef, Egypt, inferring that this new potential pathogen spreads among provinces, and *R. annulatus* is the main vector of this species in Egypt. This new species was detected in other countries such as China in *R. microplus* [54], Turkey in *Hyalomma excavatum* [55], and Pakistan in *H. anatolicum* [56]. Nevertheless, *R. annulatus* was free from piroplasms. Abdullah and her colleagues [40] confirmed our findings, stating that *R. annulatus* was *Piroplasma* negative.

## 4. Materials and Methods

### 4.1. Animals and Blood Sampling

A total of 234 animal hosts (112 cattle, 38 sheep, 28 goats, 26 buffaloes, 22 donkeys, and 8 horses) were included in a cross-sectional study using a convenience sampling strategy. These animals were collected from six provinces of Egypt (AL-Faiyum, AL-Giza, Beni-Suef, Al-Minufia, Al-Beheira, and Matruh) during the period from December 2019 to March 2020 (Figure 3, Table 4). The species, sex, age, and tick infestation of each domestic animal were recorded. Moreover, 3 mL of blood per animal was collected by sterile syringe in a sterile EDTA-Vacutainer tube. All blood samples were stored at −20 °C till molecular investigation.

### 4.2. Ticks

From the examined animals, a total of 95 ticks were collected in Eppendorf tubes containing 70% ethanol. Each Eppendorf tube was assigned to one animal host and then transferred to the laboratory for morphological identification according to the keys of Estrada-Pena and his colleagues [57]. All ticks were processed for molecular screening.

### 4.3. Molecular Investigation

#### 4.3.1. DNA Extraction

DNA was extracted from 300 µL of each blood sample using a Genomic DNA isolation Kit (Blood; GeneDireX, Taiwan, China) according to the manufacturer’s instructions. In addition, ticks were individually dipped twice in distilled water and then dried with sterile filter paper [58]. Cleaned ticks were cut longitudinally into two parts; one half was sliced up into small pieces, and another half was stored at −20 °C as a backup. Each sliced tick half was added into a sterile 1.5-mL Eppendorf containing 400 µL of lysis buffer and 10 µL of proteinase K (40 mg/µL; Simply^TM^, Taiwan, China) and incubated overnight at 65 °C. After centrifugation, the supernatant was transferred into a sterile 1.5-mL Eppendorf tube and directed to DNA extraction using a Tissue Genomic DNA isolation Kit (Tissue; GeneDireX, Taiwan, China) according to the manufacturer’s instructions. The extracted DNA from each blood and tick sample was stored at −20 °C until PCR pathogen screening.

#### 4.3.2. Screening of Pathogens DNA by Standard PCR

The primers were used targeting 969 bp of the conserved region of encoding ribosomal 18S and 500 bp of 16S RNA genes to detect *Piroplasma* [2] and Anaplasmataceae DNA [59], respectively (Table 5). PCR assays were performed using One PCR master mix™ (GeneDireX, Taiwan, China) in an automated BIO-RAD Thermal Cycler (BIO-RAD, Singapore). PCR conditions of the *Piroplasma* and Anaplasmataceae amplification were applied according to previously published methods by Dahmana and his colleagues [2] and Cardoso and his colleagues [59], respectively (Table 5). In addition, we used genus-specific primers for amplifying and sequencing *Theileria* sp. and *Babesia* sp. [60,61] (Table 5). Positive controls were *T. annulata* (MN625888), *B. bigemina* (MN625890), and *A. marginale* (MN625935) DNA extracted from cattle for PCR assays of *Piroplasma* and Anaplasmataceae, respectively, whereas distilled water was used as a negative control. Then, 1.5% agar gel stained with Red Safe electrophoresis was performed to check the amplification and then visualized by UV transilluminator. Moreover, a 100 bp DNA Ladder (GeneDireX, Taiwan, China) was used to assess the size of PCR products. A PCR Clean-Up and Gel Extraction Kit (GeneDireX, Taiwan, China) was used according to the manufacturer’s instructions to purify the PCR products of the positive samples.

#### 4.3.3. Sequencing and Phylogenetic Analyses

The purified PCR products were sequenced at the Macrogene Lab Technology, Korea. ChromasPro software (ChromasPro 1.7, Technelysium Pty Ltd., Tewantin, Australia) assembled and corrected the obtained sequences. The corrected sequences of *Piroplasma* or Anaplasmataceae were submitted to GenBank and then compared with those available in the GenBank database by NCBI BLASTn (http://blast.ncbi.nlm.nih.gov/Blast.cgi, accessed on 12 December 2021). MEGA software version X was used for multiple alignments of the obtained sequences and sequences of validated species already available in GenBank. Then, maximum-likelihood phylogenetic trees were constructed with 1000 bootstrap replications [62,63].

### 4.4. Statistical Analysis

Significant differences in the prevalence rates with *Piroplasma* and Anaplasmataceae and their risk factors such as animal species, sex, age, and tick infestation were calculated by the χ^2^ test using IBM SPSS Statistics for Windows, version 20.0 (IBM Corp., Armonk, NY, USA) at *p* < 0.05.

## 5. Conclusions

Piroplasm is still a dangerous bovine hemopathogen in Egypt, and its negative effects may increase, especially when its tick vector *R. annulatus* is found. The Anaplasmataceae prevalence rate in subclinical and carrier animals was high and needs more attention to reduce its effect on animal production. The findings of this study will provide baseline data on TBD epidemiology and tick vector management patterns, which will aid the government in establishing policies that could improve animal health security and the economy of the country. Further studies are recommended on large scales of animals and their associated arthropods (ticks, lice, and sucking flies) using species-specific primers for the detection of co-infections and novel emerging and re-emerging species and/or genotypes.

## Figures and Tables

**Figure 1 pathogens-11-01194-f001:**
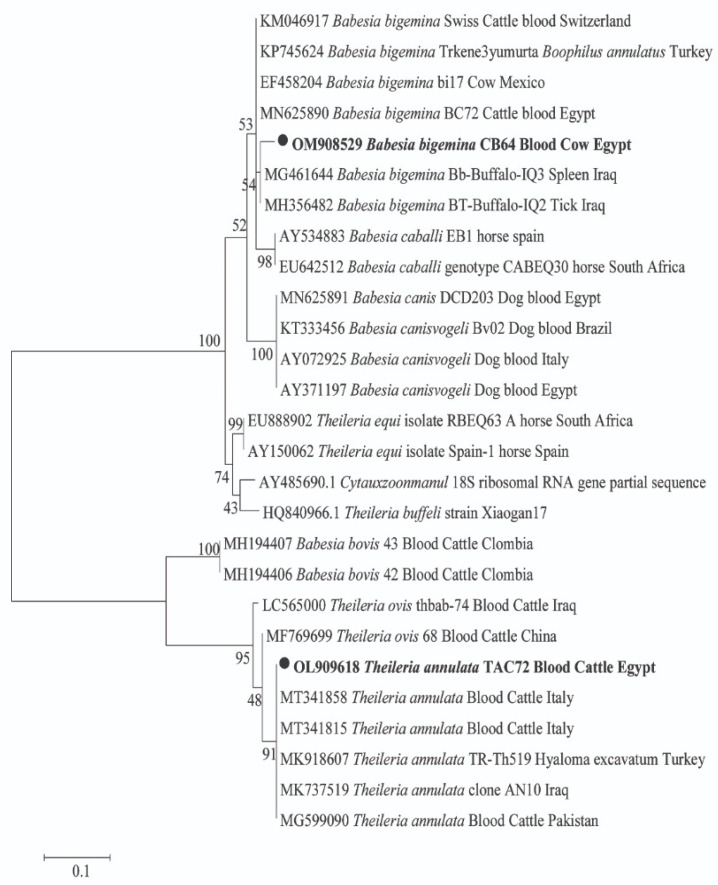
Phylogenetic tree of *Theileria* sp. and *Babesia* sp. 18S rRNA gene. The maximum likelihood method was constructed using MEGA X. Newly obtained sequences in this study are highlighted (bold).

**Figure 2 pathogens-11-01194-f002:**
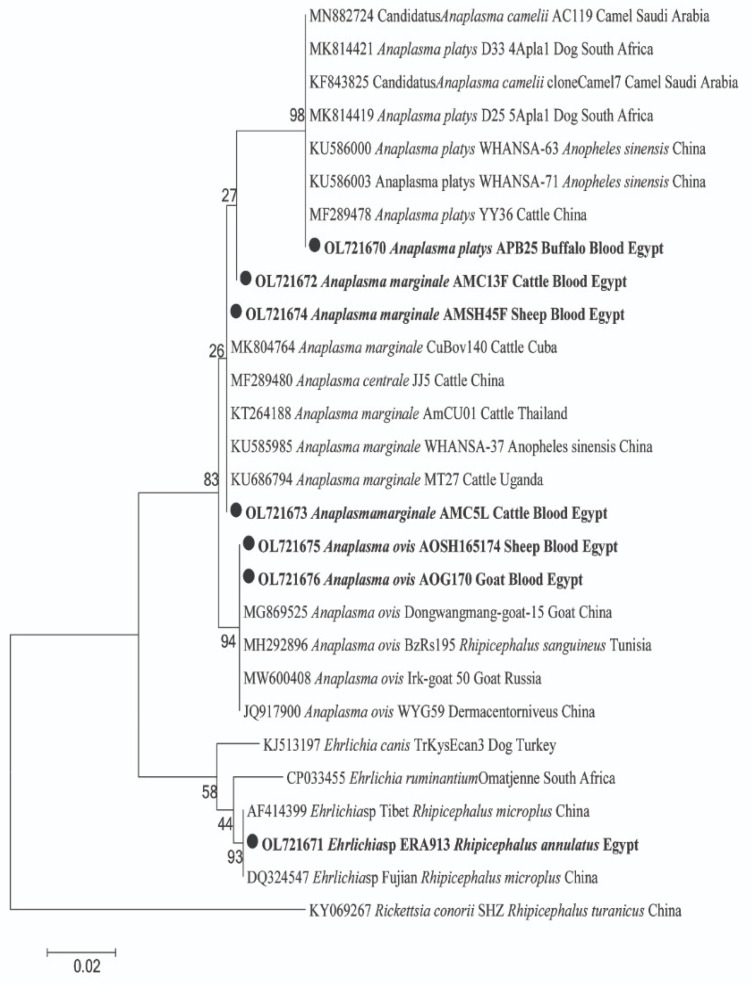
Phylogenetic tree of *Anaplasma* sp. and *Ehrlichia* sp. 16S rRNA gene. The maximum likelihood method was constructed using MEGA X. Newly obtained sequences in this study are highlighted (bold). *Rickettsia conorii* was used as an out-group.

**Figure 3 pathogens-11-01194-f003:**
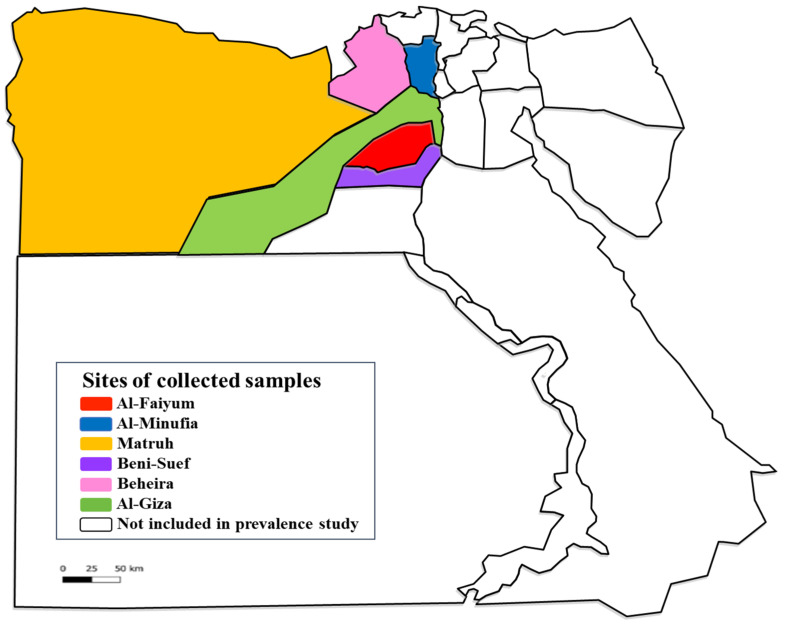
Map of Egypt presenting the provinces where the studied animals and their ticks were collected. https://en.wikipedia.org/wiki/Governorates_of_Egypt (accessed on 27 May 2022) and the picture has CC BY-SA 3.0.

**Table 1 pathogens-11-01194-t001:** Prevalence of hemopathogens (*Piroplasma* and Anaplasmataceae) in domestic animals from six provinces in Egypt from December 2019 to March 2020.

Region	Examined Animals	Hemopathogens						Overall Prevalence		χ^2^	*p* Value
		*Piroplasma*		Anaplasmataceae		Co-Infection					
		Positive	%	Positive	%	Positive	%	Positive	%		
Al-Faiyum	47	1	2.12	16	34.04	0	0	17	36.17	13.235	<0.001 **
Al-Minufia	49	0	0	2	4.08	0	0	2	4.08	-	-
Matruh	57	2	3.5	17	29.82	1	1.75	19	33.33	11.842	0.001 **
Beni-Suef	26	5	19.23	2	7.69	4	15.38	7	26.92	1.286	0.257 #
Al-Beheira	28	1	3.57	3	10.71	0	0	4	14.29	1.000	0.317 #
Al-Giza	27	1	3.7	4	14.81	0	0	5	18.52	1.800	0.180 #
Total	234	10	4.27	44	18.80	5	2.14	54	23.08	21.407	<0.001 **

** Highly significant at *p* < 0.01; # non-significant at *p* > 0.05.

**Table 2 pathogens-11-01194-t002:** Risk factors associated with the prevalence of *Piroplasma* and Anaplasmataceae.

Factor		Total Animals	*Piroplasma*		Anaplasmataceae		χ^2^	*p* Value
			Positive	%	Positive	%		
Animal species	Cattle	112	10	8.93	28	25	8.526	0.004 **
	Buffaloes	26	0	0	2	7.69	-	-
	Sheep	38	0	0	10	26.32	-	-
	Goats	28	0	0	3	10.71	-	-
	Donkeys	22	0	0	1	4.55	-	-
	Horses	8	0	0	0	0	-	-
	χ^2^			-		38.923		
	*p* value			-		<0.001 **		
Sex	Female	168	10	5.95	32	19.05	11.524	0.001 **
	Male	66	0	0	12	18.18	-	-
	χ^2^			-		0.027		
	*p* value			-		0.869 #		
Age	≤1	54	3	5.56	5	9.26	5.000	0.480 #
	>1	180	7	3.89	39	21.67	22.261	<0.001 **
	χ^2^			0.400		16.670		
	*p* value			0.527 #		<0.001 **		
Tick-infested animals	Yes	36	6	16.67	15	41.67	3.857	0.050 *
	No	198	4	2.02	29	14.64	18.939	<0.001 **
	χ^2^			11.842		12.789		
	*p* value			0.001 **		<0.001 **		

* Significant at *p* < 0.05, ** Highly significant at *p* < 0.01, and # non-significant at *p* > 0.05.

**Table 3 pathogens-11-01194-t003:** Detection of *Anaplasma* DNA in *Rhipicephalus annulatus* ticks collected from cows and buffaloes from five provinces in Egypt.

Region	Infested Animals	Anaplasmataceae Detection (*Rhipicephalus annulatus*)		
		Examined Number	Positive	%
Al-Faiyum	Cow (8)	61	42	68.85
	Buffalo (1)	10	10	100
Al-Monufia	Buffalo (4)	-	-	-
Matruh	Cow (2)	5	3	60
Beni-Suef	Cow (4)	13	10	76.92
Al-Beheira	Cow (1)	6	4	66.67
Total	20	95	69	72.63

**Table 4 pathogens-11-01194-t004:** Data of the studied animals.

Provinces	Geographic Coordinates	Animal Hosts	No. of Animals	Locations
Al-Faiyum	29°18′35.8″ N, 30°50′30.48″ E	Cattle Buffaloes Sheep Donkeys	31 6 6 4	Households
Al-Minufia	30°35′50.09″ N, 30° 59′15.48″ E	Cattle Buffaloes Sheep Goats Donkeys Horses	22 12 6 2 5 2	Households
Matruh	31°21′10.44″ N, 27°14′14.10″ E	Cattle Buffaloes Sheep Goats Donkeys Horses	11 1 17 20 6 2	Households
Beni-Suef	29°03′60.00″ N, 31°04′60.00″ E	Cattle Donkeys Horses	21 3 2	Households
Al-Beheira	30°50′53.16″ N, 30°20′36.78″ E	Cattle Buffaloes Sheep Goats Donkeys Horses	13 5 5 2 2 1	Households
Al-Giza	29°58′27.00″ N, 31°08′2.21″ E	Cattle Buffaloes Sheep Goats Donkeys Horses	14 2 4 4 2 1	Households

**Table 5 pathogens-11-01194-t005:** Oligonucleotide sequences of primers used for PCR and sequencing.

Pathogens	Targeted Gene	Primers	Tm	References
** *Piroplasma* ** *T. annulata* *B. bigemina*	18S rRNA (969 bp) 18S rRNA (360 bp) SS rRNA (689 bp)	piro18S-F1-GCGAATGGCTCATTAIAACA piro18S-F4-CACATCTAAGGAAGGCAGCA TBM-CTTCAGCACCTTGAGAGAAATC Equi-R-TGCCTTAAACTTCCTTGCGAT Bg3-TAGTTGTATTTCAGCCTCGCG Bg4-AACATCCAAGCAGCTAHTTAG	58 °C 58 °C 57 °C	Dahmana et al. (2019) [2] Qablan et al. (2013) [60] El-Naga and Barghash, et al. (2016) [61]
**Anaplasmataceae** (*Anaplasma* and *Ehrlichia*)	16S rRNA (500 bp)	ECB-CGTATTACCGCGGCTGCTGGCA ECC-AGAACGAACGCTGGCGGCAAGC	65 °C	Cardoso et al. (2016) [59]

## Data Availability

Not applicable.

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
