# Peer review of "Molecular Epidemiological Investigation of Piroplasms and Anaplasmataceae Bacteria in Egyptian Domestic Animals and Associated Ticks"

_pathogens, 2022, doi:10.3390/pathogens11101194_

Round 1

Reviewer 1 Report

The study aims to survey Piroplasma and Anaplama in livestock across 6 provinces in Egypt using molecular techniques which could potentially detect those pathogens in animals with subclinical symptoms or are carriers. 

The introduction contains all information required to understand why authors proposed to survey these pathogens in livestocks. However, the writing in the result and discussion sections needs to be revised.

Comments 

Majors;

Table 2, there is a misunderstanding the way it presents for ticks. After reading the result section, the meaning for tick in the table is actually an animal with tick infestation and the author did not present any pathogen detection on ticks in this table. The author should reword “tick” to something like “animal with tick infestation” to clarify that the data in the table are Piroplasma- or Anaplasma-positive animals with or without tick infestation. 

127-153, suggest adding another table with nucleotide sequencing identity to reference sequences for all sequences generated in this study. The identity matrix can be obtained from “compute pairwise distances” in MEGA or sequence identity matrix  in BioEdit.

166-175, please add more information on how ticks were processed, pooled? or Were they assayed individually? Were all Anaplasma-positive ticks infected with Ehrlichia sp.? Were all sequences the same? Since there was only one sequence in the phylogenetic tree therefore it could be assumed that there was only one Ehrlichia species in all Anaplasma-positive ticks. Please clarify and add more information in the result part as well as in the method section of how ticks were processed (pool or individual).

Minors;

101, what kind of domestic animals are free from piroplasmosis? Please define. Or revise the statement such as Piroplasmosis only found/detected in cattles… whereas anaplasmosis…

103, please cite Table 2 after Anaplasma DNA so that the reader can follow the story.

105, missing % after 26.32

133 and 135, 361/364 “bp”?

149, what is TB? 

166, R. annulatus was mentioned for the first time so the genus should be spelled out.

218, please recheck the overall infection rate in animals for anaplasmosis. It should be lower than 23.08%. 

Author Response

Reviewer 1:

The study aims to survey Piroplasma and Anaplasma in livestock across 6 provinces in Egypt using molecular techniques which could potentially detect those pathogens in animals with subclinical symptoms or are carriers. 

The introduction contains all information required to understand why authors proposed to survey these pathogens in livestock. However, the writing in the result and discussion sections needs to be revised.

Author’s response:

Thanks for this valuable comment. The results and discussion have been edited.

Majors;

Table 2, there is a misunderstanding the way it presents for ticks. After reading the result section, the meaning for tick in the table is actually an animal with tick infestation and the author did not present any pathogen detection on ticks in this table. The author should reword “tick” to something like “animal with tick infestation” to clarify that the data in the table are Piroplasma- or Anaplasma-positive animals with or without tick infestation. 

Author’s response:

We thank the Reviewer for this valuable comment. The item “Ticks” in table 2 has been replaced by “Tick-infested animals”

127-153, suggest adding another table with nucleotide sequencing identity to reference sequences for all sequences generated in this study. The identity matrix can be obtained from “compute pairwise distances” in MEGA or sequence identity matrix  in BioEdit.

Author’s response:

The matrixes for both pathogens have been built, but I have a suggestion to put this matrix as a supplementary data, if the reviewer wants.

Piroplasma matrix

Anaplasma Matrix

166-175, please add more information on how ticks were processed, pooled? or Were they assayed individually? Were all Anaplasma-positive ticks infected with Ehrlichia sp.? Were all sequences the same? Since there was only one sequence in the phylogenetic tree therefore it could be assumed that there was only one Ehrlichia species in all Anaplasma-positive ticks. Please clarify and add more information in the result part as well as in the method section of how ticks were processed (pool or individual).

Author’s response:

In Line 329, we mentioned that ticks processed individually for the molecular screening. We didn’t process pooling ticks. However, more details have been added (Line 337).

For Anaplasmataceae sequencing, we selected the positive samples with strong obvious bands representing all investigated animals and provinces to be sequenced. The obtained good-quality sequences were identified as Ehrlichia sp. and no other Anaplasma sp. were detected, while the bad-quality sequences were excluded. Further studies are needed using specific primers for both species (Anaplasma and Ehrlichia) to detect the possibility of co-infection. The sentence has been edited to be more clear (Line 185-186).

Minors;

101, what kind of domestic animals are free from piroplasmosis? Please define. Or revise the statement such as Piroplasmosis only found/detected in cattles… whereas anaplasmosis…

Author’s response:

The sentence has been modified (Line 107).

103, please cite Table 2 after Anaplasma DNA so that the reader can follow the story.

Author’s response:

The table has been cited (Line 110).

105, missing % after 26.32

Author’s response:

The percent has been added (Line 110).

133 and 135, 361/364 “bp”?

Author’s response:

It is the comparison between our sequences base pair with the references in GenBank.

149, what is TB? 

Author’s response:

It means Tick-borne pathogens, the sentence has been corrected (Line 161).

166, R. annulatus was mentioned for the first time so the genus should be spelled out.

Author’s response:

The name of genus has been spelled out (Line 180).

218, please recheck the overall infection rate in animals for anaplasmosis. It should be lower than 23.08%. 

Author’s response:

We thank the Reviewer for this valuable remark. The infection rate of anaplasmosis has been corrected (Line 238).

Reviewer 2 Report

Good proofreading will help polish the language of the manuscript. (For example: line 299: consider adding the word “one” before “individual tick ….”. Another example is line 324: you said “to purified” and it should be “to purify…).

Methods

Why the animal numbers were quite different and too small in some cases (8 horses)?

Line 309, the authors cannot use the term “the primers were designed” because they did not design anything, rather they used published primer sequences. They can say “we used primers targeting ……..”.

Line 315, insert the word “genus” so the sentence reads as “we used genus-specific primers …..”

Line 319, add “%” after 1.5 because the gel is described as 1.5% gel … 

Results

Figures 1 and 2 do not look good in terms of the presentation quality (journal publication level). The authors need to re-produce them and make sure that the font (size and type) and line width are the same in both figures and can be read easily. The bootstrap values were faint and hard to read. One author should make the two trees and apply the same production rules. Also, where are the out-groups in both of the phylogenetic trees? Bootstrap values less than 50 or even 75 should not be presented on the tree.

Authors are using the term “infected with” in the manuscript although in their study they only detected the presence of DNA of the microorganism. It is more accurate to talk about the presence of DNA in the blood of the host or the tick.  

Discussion

In several places, the authors used the term “novel sequence/s”. What type of novelty they are talking about? Explain.  

Line 133, the authors used the term “new genotype” which does not seem fitting. Did they compare the sequence with an already known sequence from Egypt and found it new?.

The discussion in it its current format is too local and the paper should be published in a local Egyptian journal. International readers need some take-home messages that relate to the big picture in the world. Therefore, the authors need to improve the discussion by adding points showing links between the findings of their paper and global issues such as worldwide animal trade, movement of animals within the country and between neighboring countries, zoonotic risk (when applicable), the introduction of pathogens to previously disease-free areas, and the contribution to global animal disease surveillance among other issues.

The authors need to talk about the limitations of this study and should give some recommendations for future researchers who might conduct similar work in order to improve their findings.

Author Response

Revision Letter

Dear Reviewer,

We are grateful for the valuable comments of our manuscript (pathogens-1901709) entitled "Piroplasmosis and anaplasmosis in Egyptian domestic animals: A molecular epidemiological investigation". We would like to inform you that reviewers’ comments have been made. The following are authors' responses point by point to all comments.

Regards,

Hend Abdullah

Good proofreading will help polish the language of the manuscript. (For example: line 299: consider adding the word “one” before “individual tick ….”. Another example is line 324: you said “to purified” and it should be “to purify…).

Author’s response:

Thanks for your valuable observations; however, the manuscript was edited by the official English language agency. All requested corrections have been done (Line 329, 337 and 356).

Methods

Why the animal numbers were quite different and too small in some cases (8 horses)?

Author’s response:

This study explored Piroplamosis and Anaplasmosis in domestic animals mainly cattle and we mentioned that a convenience sampling strategy was used for finding out the prevalence of Piroplamosis and Anaplasmosis at the time of the study through a cross-sectional sampling from animals’ population. Furthermore, a few owners in the investigated provinces possessed horses.

Line 309, the authors cannot use the term “the primers were designed” because they did not design anything, rather they used published primer sequences. They can say “we used primers targeting ……..”.

Author’s response:

The sentence has been edited (Line 340).

Line 315, insert the word “genus” so the sentence reads as “we used genus-specific primers …..”

Author’s response:

The word ‘genus’ was added (Line 347).

Line 319, add “%” after 1.5 because the gel is described as 1.5% gel …

Author’s response:

The ‘%’ has been added (Line 351).

Results

Figures 1 and 2 do not look good in terms of the presentation quality (journal publication level). The authors need to re-produce them and make sure that the font (size and type) and line width are the same in both figures and can be read easily. The bootstrap values were faint and hard to read. One author should make the two trees and apply the same production rules. Also, where are the out-groups in both of the phylogenetic trees? Bootstrap values less than 50 or even 75 should not be presented on the tree.

Author’s response:

Two figures 1 & 2 have been edited according to the reviewer’s comments.

Anaplasmataceae tree, we used Rickettsia conorii as an out-group, while Piroplasma spp. tree was constructed as Babesia sp. were collected in a separate clade with 100 bootstrap, and Theileria sp. has the same condition. So, we didn’t use an out-group. 

Authors are using the term “infected with” in the manuscript although in their study they only detected the presence of DNA of the microorganism. It is more accurate to talk about the presence of DNA in the blood of the host or the tick.

Author’s response:

It is true. The manuscript has been revised and edited with “detected/detection or positive for…”.

Discussion

In several places, the authors used the term “novel sequence/s”. What type of novelty they are talking about? Explain.

Line 133, the authors used the term “new genotype” which does not seem fitting. Did they compare the sequence with an already known sequence from Egypt and found it new?.

Author’s response:

We used term “new genotype/novel sequences”, when we found that our sequences were not 100% identity with other reference sequences in GenBank. Moreover, our obtained sequences differed from other reference sequences in GenBank with one base pair or more, when using Blast analysis or multiple sequence alignment in MEGA X. Some obtained sequences were 100% similarity with one or more reference sequences in GenBank, so they didn’t call novel genotypes. However, we have added “potential” to be more accurate.

For Ehrlichia sp., we called it novel species, because it detected in R. annulata collected from cattle while others detected in other Rhipicephalus species or other tick species. Also, in the conclusion section, we have added the limitation of primers (Line 381-384).

The discussion in it its current format is too local and the paper should be published in a local Egyptian journal. International readers need some take-home messages that relate to the big picture in the world. Therefore, the authors need to improve the discussion by adding points showing links between the findings of their paper and global issues such as worldwide animal trade, movement of animals within the country and between neighboring countries, zoonotic risk (when applicable), the introduction of pathogens to previously disease-free areas, and the contribution to global animal disease surveillance among other issues.

Author’s response:

The discussion section has been edited to be fit with the reviewer’s comment.

The authors need to talk about the limitations of this study and should give some recommendations for future researchers who might conduct similar work in order to improve their findings.

Author’s response:

The limitation and recommendation have been added in the conclusion section (Line 381-384).

Reviewer 3 Report

Dear editor and authors, 

I read the MS entitled "Piroplasmosis and anaplasmosis in Egyptian domestic animals: A molecular epidemiological investigation" and I have the following comments and concerns:

General comments:

The genetical characterisation is limited, the genes 18S and 16S are commonly used for bacterial species identification, however when are involved highly related species, it was observed that are not enough for a proper identification, especially if the sequence length is short (under 500 bp) as in case of Anaplasma. In addition, the authors speak about new genotype or unknown species but based on their limited genetic analysis (only one marker), the results they present may be considered misleading. Another aspect is related to the protocol used, in the results the Anaplasma sp. prevalence is reported, but the protocol used detect both Anaplasma and Ehrlichia species.

The results itself are of local importance, and don’t bring any new information in the general epidemiology of the studied bacteria. In my opinion, for an epidemiological survey the number of samples is to low. If the authors focus on the species primary identified by 18S and 16S, additional genetical markers can be used to proper characterise their findings (for instance the unknow Ehrlichia sp. or A. platys from buffalo), and thus the importance of the study will increase. 

The discussions are limited by the importance of the results. If additional genetical characterisation is added then the discussion can be also improved. The limitation of the study are not discussed.

The writing style should be improved, several sentences (many) are to long and hard to follow.

Specific comments:

Abstract:

line 17: rewrite, is not clear, 95 ticks collected from the sampled animals? if all samples originated from the six mentioned provinces, change “in addition to” with “and”;

lines 21-23: rewrite, is hard to follow;

line 54: delete the parenthesis, is not complete if refers to rickettsial pathogens and is more misleading than helpful;

lines 91-96: this section repeat the result presented in the table 1;

Lines 100-124: similarly, this entire section repeat the result presented in the table 2; the results should be rewrite to summarize the results presented in tables;

….

sequencing and phylogenetic analysis:

3 nucleotide difference is not necessary a new genotype; I personally think that is needed more depth genetic analysis on several markers in order to speak about new genotype, in addition, 18S and 16S are useful for bacteria identification but considering the recent genetic analysis and reports are not enough. For instance, there are several A. platys -like species which has been recent genetically characterised, and even new Anaplasma species or Candidatus related to the valid species are continuously reported. Based on this I do not think that 16S sequencing is enough anymore to properly differentiate between all these Anaplasma organisms.

all 69 Ehrlichia sp. isolated in ticks were sequenced and were identical among each other and 100% with that reported in China? 

Author Response

Revision Letter

Dear Reviewer,

We are grateful for the valuable comments of our manuscript (pathogens-1901709) entitled "Piroplasmosis and anaplasmosis in Egyptian domestic animals: A molecular epidemiological investigation". We would like to inform you that reviewers’ comments have been made. The following are authors' responses point by point to all comments.

Regards,

Hend Abdullah

I read the MS entitled "Piroplasmosis and anaplasmosis in Egyptian domestic animals: A molecular epidemiological investigation" and I have the following comments and concerns:

General comments:

The genetical characterisation is limited, the genes 18S and 16S are commonly used for bacterial species identification, however when are involved highly related species, it was observed that are not enough for a proper identification, especially if the sequence length is short (under 500 bp) as in case of Anaplasma. In addition, the authors speak about new genotype or unknown species but based on their limited genetic analysis (only one marker), the results they present may be considered misleading. Another aspect is related to the protocol used, in the results the Anaplasma sp. prevalence is reported, but the protocol used detect both Anaplasma and Ehrlichia species.

Author’s response:

We used 18S and 16S RNA genes specific for Piroplasm and Anaplasmataceae, they aren’t common primers for bacterial detection. However, we have added some limitations in the conclusion section including the recommendation for using species specific primers and more genetic markers.

All manuscript has been edited by replacing ‘Anaplasma sp.’ by ‘anaplasmataecae’ to be more accurate. 

The results itself are of local importance, and don’t bring any new information in the general epidemiology of the studied bacteria. In my opinion, for an epidemiological survey the number of samples is too low. If the authors focus on the species primary identified by 18S and 16S, additional genetical markers can be used to proper characterise their findings (for instance the unknow Ehrlichia sp. or A. platys from buffalo), and thus the importance of the study will increase. 

Author’s response:

Thanks for this valuable observation. We have edited the manuscript especially discussion to make a link with the international investigations.

We have added some limitations related to the primers used.

Regarding samples, this study explored Piroplamosis and Anaplasmosis in domestic animals mainly cattle and we mentioned that a convenience sampling strategy was used for finding out the prevalence of Piroplamosis and Anaplasmosis at the time of the study through a cross-sectional sampling from animals’ population. Furthermore, a few owners in the investigated provinces possessed horses.

The discussions are limited by the importance of the results. If additional genetical characterisation is added then the discussion can be also improved. The limitation of the study are not discussed.

Author’s response:

The limitation and recommendation have been added in the conclusion section (Line 381-384).

“Further studies are recommended on large scales of animals and their associated arthropods (ticks, lice, and sucking flies) using species-specific primers for detection of co-infection and novel emerging and re-emerging genotypes.”

The writing style should be improved, several sentences (many) are to long and hard to follow.

Author’s response:

The manuscript was edited by the official English language agency and the manuscript has been revised again.

Specific comments:

Abstract:

line 17: rewrite, is not clear, 95 ticks collected from the sampled animals? if all samples originated from the six mentioned provinces, change “in addition to” with “and”;

Author’s response:

It has been added (Line 17).

lines 21-23: rewrite, is hard to follow;

Author’s response:

The sentence has been edited (Line 22 to 24).

line 54: delete the parenthesis, is not complete if refers to rickettsial pathogens and is more misleading than helpful;

Author’s response:

The sentence has been edited by replacing “The rickettsial pathogen” with “Family Anaplasmataceae” (Line 57 and 58)

lines 91-96: this section repeat the result presented in the table 1;

Author’s response:

The repeated results were deleted (Line 87 to 102).

Lines 100-124: similarly, this entire section repeat the result presented in the table 2; the results should be rewrite to summarize the results presented in tables;

Author’s response:

The repeated results were deleted (Line 107 to 134).

sequencing and phylogenetic analysis:

3 nucleotide difference is not necessary a new genotype; I personally think that is needed more depth genetic analysis on several markers in order to speak about new genotype, in addition, 18S and 16S are useful for bacteria identification but considering the recent genetic analysis and reports are not enough. For instance, there are several A. platys -like species which has been recent genetically characterised, and even new Anaplasma species or Candidatus related to the valid species are continuously reported. Based on this I do not think that 16S sequencing is enough anymore to properly differentiate between all these Anaplasma organisms.

 Author’s response:

We used term “new genotype/novel sequences”, when we found that our sequences were not 100% identity with other reference sequences in GenBank. Moreover, our obtained sequences differed from other reference sequences in GenBank with one base pair or more, when using Blast analysis or multiple sequence alignment in MEGA X. Some obtained sequences were 100% similarity with one or more reference sequences in GenBank, so they didn’t call novel genotypes. However, we have added “potential” to be more accurate.

Furthermore, we have added some limitations including the recommendation for using species specific primers and more genetic markers.

all 69 Ehrlichia sp. isolated in ticks were sequenced and were identical among each other and 100% with that reported in China? 

Author’s response:

For Anaplasmataceae sequencing, we selected the positive samples with strong obvious bands representing all investigated animals and provinces to be sequenced. The obtained good-quality sequences were identified as Ehrlichia sp. and no other Anaplasma sp. were detected, while the bad-quality sequences were excluded. Further studies are needed using specific primers for both species (Anaplasma and Ehrlichia) to detect the possibility of co-infection. The sentence has been edited to be more clear (Line 185-186).

Reviewer 4 Report

Sobhy Abdel-Shafy et al submitted a manuscript describing detection of piroplasma (babesia and theileria) and anaplasma in sera of livestock and ticks feeding on livestock. The work is done following the standard procedures and its results are in a good concordance with the previously published results (which also means that they do not bring any super new information). Despite I think that the work deserves to be published at least to add more information about prevalence of these pathogens in Egypt.

Before being accepted I think that authors should improve stylistic quality of English used in the manuscript. I am not a native speaker but I found some of the paragraphs too long and therefore too hard to follow. I also noticed overusing of some excessive and overused words (e.g., whereas).

Apart this I also have several minor comments:   

1) Coinfections by babesia and theileria are not mentioned in the abstract only piroplasma as one group.

2) Lines 49-51: Babesia causes ... disease prevention. - The sentence has to be reformulated. Babesia does not cause disease prevention it requires disease prevention or it causes loses connected to diseases prevention.

3) Line 86 - except instead of expect

4) Figure 3 - add the scale to the map

Author Response

Sobhy Abdel-Shafy et al submitted a manuscript describing detection of piroplasma (babesia and theileria) and anaplasma in sera of livestock and ticks feeding on livestock. The work is done following the standard procedures and its results are in a good concordance with the previously published results (which also means that they do not bring any super new information). Despite I think that the work deserves to be published at least to add more information about prevalence of these pathogens in Egypt.

Before being accepted I think that authors should improve stylistic quality of English used in the manuscript. I am not a native speaker but I found some of the paragraphs too long and therefore too hard to follow. I also noticed overusing of some excessive and overused words (e.g., whereas).

Author’s response:

The manuscript was edited by the official English language agency and the manuscript has been revised again.

Apart this I also have several minor comments:   

1) Co-infections by babesia and theileria are not mentioned in the abstract only piroplasma as one group.

 Author’s response:

We mentioned total co-infection in the abstract; however, we added the details of co-infection in the abstract (Line 24, 25 and 26).

2) Lines 49-51: Babesia causes ... disease prevention. - The sentence has to be reformulated. Babesia does not cause disease prevention it requires disease prevention or it causes loses connected to diseases prevention.

 Author’s response:

The sentence has been corrected (Line 54).

3) Line 86 - except instead of expect

 Author’s response:

The word has been corrected (Line 91).

But the sentence has been deleted to summarize the results (another reviewer request).

4) Figure 3 - add the scale to the map

 Author’s response:

The scale of the map has been added.

Round 2

Reviewer 2 Report

I would like to thank the authors for the changes they made to the manuscript; now it looks better. They need to consider the following points:

Line 129: change it from “a new potential genotype” to “a potential new genotype” because the word “potential” was added to modify the meaning of the word “new” and not the word genotype.

Line 130: change it to “Moreover, a potential new ……”.

Line 137: change it to “Another potential novel ….”.

Line 271: change it to “Regarding pathogens’ DNA detection in ticks ….”

Line 280: add the word “and” before the word Pakistan.

Author Response

I would like to thank the authors for the changes they made to the manuscript; now it looks better. They need to consider the following points:

Author response:

Thanks for your valuable response. Kindly, find our response point by point.

Regards,

Hend Abdullah

Line 129: change it from “a new potential genotype” to “a potential new genotype” because the word “potential” was added to modify the meaning of the word “new” and not the word genotype.

Author response:

The word position has been changed (Line 144).

Line 130: change it to “Moreover, a potential new ……”.

Author response:

The word position has been changed (Line 145).

Line 137: change it to “Another potential novel ….”.

Author response:

The word position has been changed (Line 153).

Line 271: change it to “Regarding pathogens’ DNA detection in ticks ….”

Author response:

The sentence has been edited (Line 290).

Line 280: add the word “and” before the word Pakistan.

Author response:

The sentence has been edited (Line 299).
